# Application of Polyethylene Glycol-Based Flame-Retardant Phase Change Materials in the Thermal Management of Lithium-Ion Batteries

**DOI:** 10.3390/polym15224450

**Published:** 2023-11-17

**Authors:** Yan Gong, Jiaxin Zhang, Yin Chen, Dongxu Ouyang, Mingyi Chen

**Affiliations:** 1School of the Environment and Safety Engineering, Jiangsu University, Zhenjiang 212013, China; 2222209093@stmail.ujs.edu.cn (Y.G.); 18871550611@163.com (J.Z.); 1000005956@ujs.edu.cn (Y.C.); 2School of Emergency Management, Jiangsu University, Zhenjiang 212013, China; 3School of Safety Science and Engineering, Nanjing Tech University, Nanjing 210009, China

**Keywords:** bio-based flame retardants, flame retardant phase change materials, PEG, thermal properties, battery thermal management

## Abstract

Composite phase change materials commonly exhibit drawbacks, such as low thermal conductivity, flammability, and potential leakage. This study focuses on the development of a novel flame-retardant phase change material (RPCM). The material’s characteristics and its application in the thermal management of lithium-ion batteries are investigated. Polyethylene glycol (PEG) serves as the medium for phase change; expanded graphite (EG) and multi-walled carbon nanotubes (MWCNT) are incorporated. Moreover, an intumescent flame retardant (IFR) system based on ammonium polyphosphate (APP) is constructed, aided by the inclusion of bio-based flame-retardant chitosan (CS) and barium phytate (PA-Ba), which can improve the flame retardancy of the material. Experimental results demonstrate that the RPCM, containing 15% IFR content, exhibits outstanding flame retardancy, achieving a V-0 flame retardant rating in vertical combustion tests. Moreover, the material exhibits excellent thermomechanical properties and thermal stability. Notably, the material’s thermal conductivity is 558% higher than that of pure PEG. After 2C and 3C high-rate discharge cycles, the highest temperature reached by the battery module cooled with RPCM is 18.71 °C lower than that of natural air-cooling; the material significantly reduces the temperature difference within the module by 62.7%, which achieves efficient and safe thermal management.

## 1. Introduction

In recent years, the global issues of energy crisis and environmental pollution have emerged due to the escalating demand for fossil fuels [1]. Electric vehicles, one of the new energy sources, can reduce the consumption of fossil fuels to alleviate these problems by replacing conventional fuel vehicles [2,3,4]. Lithium-ion batteries (LIBs) are extensively used in electric vehicles and energy storage devices due to their high energy and power densities, long cycle life, and minimal self-discharge rate [5,6,7]. The global market for LIBs has been rapidly growing in response to the increasing demand for electric vehicles.

However, it has been observed that during the charging and discharging processes, a substantial amount of heat is generated in the battery modules and packs due to the ohmic heating effect and electrochemical reactions. This heat generation raises the battery temperature and accelerates the degradation of battery performance. With the continuous improvement of LIB energy density and the rapid development of fast charging technology, heat generation is expected to increase further [8]. Therefore, it is crucial to control the operating temperature of LIBs within an optimal range of 20 to 55 °C [9], with a safe temperature limit of 60 °C during use [10] and a maximum temperature difference of no more than 5 °C within the battery module [11]. To achieve this, an effective battery thermal management system (BTMS) is required to provide high energy density and long battery life. The BTMS plays a vital role in ensuring the performance, safety, and lifespan of LIBs [12], as it effectively regulates the battery temperature and prevents thermal runaway.

The current research on battery cooling modes can be categorized based on different cooling mediums, including air cooling [13,14], liquid cooling [15,16], heat pipe cooling [17], phase change materials (PCMs) cooling [18,19,20], and combined cooling strategies [21]. Active cooling systems, such as those employing pumps or fans, consume additional power. In contrast, passive cooling systems like PCM cooling dissipate battery surface temperature without consuming additional energy. PCM cooling technology is extensively employed in the BTMS due to its high energy density and compact nature. In PCM cooling, organic PCMs are the preferred choice. Compared to inorganic PCMs, organic PCMs offer several advantages: high thermal energy storage capacity, low or no subcooling, and no metal corrosion. For example, PEG exhibits a high latent heat of phase transition, a wide range of temperature transitions, excellent thermal and chemical stability, affordability, non-toxicity, and non-corrosiveness [22].

However, organic PCMs have weaknesses, like poor thermal conductivity, susceptibility to leakage, and flammability, significantly restricting their applications. Organic PCMs’ poor thermal conductivity can cause a significant temperature rise due to heat accumulation during the PCM’s heat absorption process, posing a risk of overheating to lithium-ion batteries. By incorporating thermally conductive additives such as expanded graphite (EG) [23,24], carbon fiber [25], metal foam [26,27], metal mesh [19,28], carbon nanotubes (CNTs) [29], and graphene [30] into PCMs, the materials’ thermal conductivity can be increased, facilitating the absorption of heat released by batteries and preventing heat accumulation in the battery module. Among these, EG and CNT are extensively employed. The CPCM was prepared by incorporating different proportions of graphene, CNTs, EG, and paraffin [31]. The addition of carbon additives with different dimensions formed a three-dimensional heat transfer network in the PCM, resulting in a thermal conductivity increase to 5.1 W/m·K at 25 °C, exhibiting high thermal conductivity. Leakage tests showed that the weight and morphology of the material remained almost unchanged after heating at 60 °C for 120 min, indicating high leak-proof properties endowed by the additives. A highly thermally conductive and shape-stable composite material was developed by impregnating single-walled carbon nanotubes (SWCNTs) into PEG [29]. The influence of different loadings of SWCNTs on the stability and heat transfer performance of PEG was investigated. The experiments demonstrated that SWCNTs exhibited excellent adsorption capacity and leak-proof characteristics, maintaining their shape, and preventing any PEG leakage even after 400 melting and freezing cycles. Compared to pure PEG, an increase in SWCNTs content created more thermal pathways, resulting in a 1329% enhancement in thermal conductivity when the mass fraction of SWCNTs reached 10%.

Furthermore, when a battery experiences thermal runaway, the PCM surrounding the battery can easily ignite and exacerbate the propagation of thermal runaway [32]. To mitigate this issue, many experiments have been conducted to enhance the flame retardancy of PCMs by adding flame retardants [33,34]. Halogen-free flame retardants, especially intumescent flame retardants (IFRs), have gained wide usage in many industries due to their high efficiency, low smoke production, and non-toxicity [20]. APP is a commonly used intumescent flame retardant that generates polyphosphoric acid upon thermal decomposition. This acid leads to the formation of a carbonized film on the surface of organic materials, which acts as a barrier against combustion. APP can also produce non-combustible gases such as nitrogen and ammonia, diluting the oxygen supply and reducing the flammability. A flame retardant PCM for battery modules using APP and red phosphorus (RP) was developed [35], and the experimenters conducted a comprehensive investigation on the flame-retardant properties of the materials with varying ratios of flame retardants and found that a ratio of 23/10 exhibited the best flame-retardant properties. In response to the need for sustainable development and the ban on traditional halogenated flame retardants, there is increasing effort across various industries to develop eco-friendly and sustainable flame-retardant materials.

There has been considerable interest in bio-based flame retardants, including Chitosan (CS) [36,37,38], Phytic acid (PA) [39,40], and lignin. These Bio-based flame retardants have been successfully applied for flame retardant modification of various substrates, including polylactic acid (PLA). Nevertheless, their application in phase change materials has been relatively limited. Among them, CS is particularly noteworthy due to its safety, environmental friendliness, and natural abundance. CS is often used in combination with other components, for instance, APP and aluminum hypophosphite, rather than as a standalone flame retardant. The chemical structure of chitosan, which is rich in hydroxyl and amine groups, makes it suitable as a carbonization agent in intumescent flame retardants. An environmentally friendly flame retardant was developed for unsaturated polyester resins; this was achieved by depositing intumescent flame retardant structures of CS and APP onto diatomaceous earth particles using a layer-by-layer assembly method [38]. An inorganic-organic nanohybrid flame retardant was prepared by combining halloysite nanotubes (HNTs), CS, and Fe_3_O_4_ [41]. The aim was to improve the thermal stability and enhance the flame retardancy of EP. CS acted as a carbon-forming agent, and the addition of Fe_3_O_4_ nanoparticles further promoted the carbonization of chitosan in EP. The combined HNT@chitosan@Fe_3_O_4_ system effectively inhibited the decomposition of EP and the release of toxic gases. Furthermore, better flame-retardant effects can be achieved when CS is compounded with phosphorus-containing compounds [42]. PA, with its six phosphate groups and high phosphorus content (up to 28%), can serve as an acid source. Phosphorus compounds facilitate the cross-linking reaction of IFR. The presence of phytates can also trap free radicals, thereby achieving gas-phase flame retardancy. An IFR system was developed, which consisted of PA and CS, with metal ions as synergists [40]. The inclusion of phytate metal salts synergistically enhanced the IFR system and accelerated the formation of non-flammable gases. The flame retardancy of PLA was further enhanced by incorporating nickel phytate (PA-Ni) into PLA containing IFR [39]. The results indicated that PA-Ni can synergistically inhibit the combustion of PLA in combination with IFR, promote the formation of dense char residue, and act as a radical scavenger to eliminate the radicals generated during the combustion process. In conclusion, the incorporation of CS and phytate salts in PCM can effectively enhance the flame retardancy of phase change materials.

Traditional flame retardants are derived from unsustainable sources and typically require a minimum additive content of at least 20% in PCM to achieve effective flame retardancy, which leads to a significant reduction in the latent heat capacity of the CPCM [43] and the incorporation of substantial quantities of flame retardants can lead to mechanical property degradation and thermophysical properties experience a decline. [44]. In contrast, bio-based flame retardants, as relatively new alternatives, have demonstrated good flame-retardant performance while being more sustainable and environmentally friendly. In this study, CS was used as a carbonizing agent, and PA-Ba was used as a synergistic agent to create an environmentally friendly IFR system based on APP. This system allowed for a reduction in the IFR content in CPCM to 15% by mass while still maintaining excellent flame retardancy. Moreover, the IFR not only enhanced the flame-retardant capability of CPCM but also improved the material’s high-temperature thermal stability. Under battery module charging and discharging operations, the resulting flame-retardant PCM exhibited the same excellent heat-diffusion capability.

## 2. Material Preparation and Experiment

### 2.1. Materials

PEG (M_w_ = 4000) obtained from Heowns Opde Technologies Co., Ltd. (Tianjin, China) served as the primary PCM in this study. EG was supplied by TengShengDa Carbon Graphite Co., Ltd. (Qingdao, China) with 99% purity. APP was purchased from Macklin Biochemical Technology Co., Ltd. (Shanghai, China). MWCNT was purchased from JiaZhaoYe New Material Co., Ltd. (Shenzhen, China). CS and barium carbonate were purchased from Bide Pharmatech Co., Ltd. (Shanghai, China). PA 50% aqueous solution was purchased from Heowns Opde Technologies Co., Ltd. (Tianjin, China). Deionized water was obtained from the laboratory. PA-Ba was prepared from the laboratory.

### 2.2. Preparation of Barium Phytate

PA and barium carbonate reacted in deionized water to prepare barium phytate, as shown in Figure 1. An appropriate amount of deionized water was added to a container, and barium carbonate was suspended in the water. A certain amount of phytic acid solution was added to another portion of deionized water and stirred until the phytic acid dissolved. The molar ratio of barium carbonate to phytic acid was determined as 3:1 based on the stoichiometric ratio. The phytic acid solution is slowly added to the barium carbonate solution using a separatory funnel while stirring. The reaction was carried out at 100 °C for 3 h, resulting in the formation of a white precipitate. The precipitate was washed with deionized water, centrifuged, and then dried at 75 °C for 10 h to obtain white PA-Ba powder.

### 2.3. Preparation of Flame Retardant CPCMs

Figure 2 illustrates the preparation process of RPCM. Employing a melt blending method. A bio-based IFR was chosen to prepare the flame retardant CPCMs, comprising APP, CS, and PA-Ba, with mass fractions of 40 wt%, 40 wt%, and 20 wt%, respectively. Initially, a certain amount of PEG was added to a beaker and heated in a thermostatic magnetic stirrer at 90 °C. Once the PEG melted, EG, MWCNT, and IFR powder (consisting of APP, CS, and PA-Ba) were gradually added to the beaker and thoroughly stirred until a uniform mixture was obtained. The materials were then cooled to room temperature to obtain CPCM powder, which could be pressed into specific shapes using custom molds. Table 1 presents the composition ratios and material nomenclature of the flame retardant CPCMs.

### 2.4. Performance Testing and Characterization of RPCM

An X-ray diffractometer (XRD) (D8 ADVANCE, Bruker, Mannheim, Germany) was employed to analyze the crystal structure and diffract the sample material. The test utilized Cu Kα radiation (wavelength: 1.5406 nm), with a step size of 0.02°, a scanning angle ranging from 10° to 80°, and a scanning speed of 7°/min. This analysis aimed to determine whether any new materials were formed during the physical melt blending process.

The Regulus-8100 Field emission scanning electron microscopy (SEM) (Hitachi, Tokyo, Japan) was utilized to examine the microstructure and composition distribution of the PCM.

A Q20 Differential Scanning Calorimeter (DSC) from TA Instruments (New Castle, DE, USA) was utilized to study the phase transition properties and latent heat of CPCMs. The sample was heated from the initial equilibrium temperature of 0 °C to 100 °C at a rate of 10 °C/min under a nitrogen atmosphere. Subsequently, it was cooled back to 0 °C at the same rate of 10 °C/min after being held at 100 °C for 5 min.

A multifunctional thermal conductivity tester (Dre-III, Xiangtan Xiangyi Instruments Co., Ltd., Xiangtan, China), with a measurement accuracy of ±3%, was used to measure the thermal conductivity of the material. The material was compacted into two disc-shaped samples with a thickness of approximately 8 mm and a diameter of 40 mm for testing. To minimize contact thermal resistance, the surfaces of the cylinders were polished. During the testing, the samples were kept stationary to ensure accurate data and multiple measurements were taken to obtain an average value.

To evaluate the leakage resistance of CPCMs, a mass loss test was conducted. During the test, samples were placed in an oven set at 75 °C. The filter paper was positioned beneath the sample surface to absorb any potential leakage. The samples were weighed every hour, a total of six times, to measure the mass loss of the material and assess its shape stability. The initial mass of the sample was denoted as $M_{O}$, while the remaining mass after n hours of heating was denoted as $M_{n}$. The leakage rate ($L_{n}$) after n hours of heating was calculated using the following equation: $L_{n}(\mathrm{wt}.\%)=\left(M_{O}{-M}_{n}\right)/M_{O}\times 100\%$. 

The thermal stability and thermal degradation of CPCMs were investigated using thermogravimetric experiments conducted on a Q500 thermogravimetric analyzer (TGA) from TA Instruments, Inc., USA. The material was heated from an initial temperature of 30 °C to 800 °C at a rate of 20 °C/min in a nitrogen atmosphere.

The flame retardancy of the samples was evaluated using the UL Vertical Flame Test conducted by a CZF-1 Vertical Flame Tester. The size of the samples used in the test was 125 mm × 13 mm × 8 mm. During the test; the samples were positioned vertically and exposed to the ignition source twice, with each exposure lasting 10 s. Flame lengths of 20 ± 1 mm were placed in the center of the samples, underneath them.

The test results are classified into one of the following categories: V-0, V-1, V-2, or NR (not rated). The V-0 rating indicates the highest level of flame retardancy, while V-1 and V-2 represent lower levels of flame retardancy. The NR classification means that the sample did not receive a flame-retardant rating.

### 2.5. Experimental Setup

The experiment utilized a Sanyo NCR18650GA LIB, which has a capacity of 3200 mA and a rated voltage of 3.7 V. Upon purchase, the new battery undergoes a cycle of charging and discharging, followed by a 24 h stabilization period before usage. Figure 3 illustrates the configuration of the battery thermal management platform, which consists of four main components: the charging and discharging system, the data acquisition module, the computer terminal, and the thermostat box.

In the experimental setup, the battery thermal management module is placed in a constant temperature box set at 25 °C. Cycle testing was performed using a battery charge–discharge system (CT4008, Shenzhen Xinhua Technology Co., Ltd., Shenzhen, China). Temperature data of the battery surface is collected using a T-type thermocouple and an Agilent data acquisition module (7018, Flow layer, Hefei, China).

The specific working conditions for thermal management are as follows: power discharge at rates of 1C, 2C, and 3C until the voltage drops to 2.75 V, followed by a 5 min rest period. The battery is then subjected to constant current and constant voltage charging until it reaches a charging current of 0.05 A, completing one cycle. The thermocouple is positioned in the middle of the battery, and the battery module is arranged symmetrically. The recorded data is used to determine the temperature and thermal gradient within the battery cycle. The temperature difference represents the highest variation in temperature between different cells in the battery. Based on the collected data, the practical effectiveness of RPCM in battery thermal management is analyzed.

## 3. Results and Discussions

### 3.1. Microscopic Characterization and Thermophysical Properties

The XRD results are displayed in Figure 4. PEG exhibits distinct diffraction peaks at 19.24° and 23.39°, while EG shows a prominent peak at 26.42°. MWCNT displays a diffraction peak at 25.96°, and IFR exhibits peaks at 14.68° and 15.52°. Notably, CPCM1 shares the same diffraction angles as PEG, EG, MECNT, and IFR without any additional peaks. Similarly, RPCM1-RPCM5 also exhibits diffraction peaks at the same 2-Theta values. However, the peaks at 19.24° and 23.39° weaken with a decrease in the proportion of PEG; this can be ascribed to the dispersion of granular IFR in PEG, which affects the crystallinity of PEG. Importantly, no new diffraction peaks are observed for any of the CPCMs, indicating that the synthesis of PCMs involves solely physical mixing rather than chemical reactions.

The morphology and microstructures of EG, SWCNT, PA-Ba, pure PEG, and their composite materials were characterized using SEM. Figure 5a illustrates the worm-like structure of EG with a rough surface featuring loose and irregular network-like pores and graphite flakes. These pores and flakes increase the specific surface area, providing space for the adsorption of PEG and IFR. Figure 5b shows the irregular particle structure of PA-Ba, which can be adsorbed into the microcavities of EG. Figure 5c exhibits MWCNTs arranged in a tubular network structure, with randomly distributed nanotubes and a high aspect ratio, facilitating the formation of thermal conductive pathways. The surface of pure PEG appears smooth and uniform in Figure 5d. Figure 5e–j presents the images of various CPCM samples, where the rough surface of CPCM is attributed to the successful adsorption of PEG into the gaps of EG. Additionally, entangled clusters of MWCNTs with different lengths and diameters can be observed in the material; this is mainly due to their significant surface area and powerful van der Waals forces, which result in poor dispersion and hinder the formation of a thermal conductive network.

The DSC curve of CPCMs is shown in Figure 6. Energy storage performance parameters of different composite PCMs were obtained and are presented in Table 2. The peak phase change temperature is defined as $T_{s\text{-}l}$. According to the data in the table, CPCM1 has a latent heat of 166.51 J/g and a phase change temperature of 54.62 °C. When 30% IFR is added, the latent heat decreases to 119.45 J/g, but it is still at a relatively high level compared to other similar PCMs [45,46], and the phase change temperature is 51.09 °C. It can be observed that the latent heat of CPCMs gradually decreases, and the phase change temperature slightly decreases. However, since the CPCM does not undergo any chemical reactions, the melting point does not change significantly. The latent heat of CPCMs is mainly determined by PEG. Enthalpy analysis indicates that a greater decrease in the mass ratio of PEG leads to a greater loss of enthalpy, resulting in a decrease in the material’s latent heat. As a result, RPCM1 and RPCM2 possess relatively high latent heat. Fire-resistant particles affect the molecular motion during the phase change process, leading to a reduction in latent heat. Therefore, adding too many flame-retardant particles will affect the energy storage performance of the material. When preparing the material, the content of functional carriers in the composite material should be considered.

As shown in Figure 7, the thermal conductivity test results of CPCMs indicate that thermal conductivity is primarily influenced by the thermal additives EG and MWCNT. The thermal conductivity of pure PEG is only 0.31 W/m·K. However, when 5% EG and 2% MWCNT are added, the thermal conductivity of CPCMs remains above 1.89 W/m·K, representing a minimum of 6 times higher than that of PEG. When 30% IFR is added, CPCM6 exhibits the lowest thermal conductivity of 1.89 W/m·K. Overall, the thermal conductivity values of all the composite phase change materials meet the requirements for thermal management.

### 3.2. Thermal Stability, Quality Stability, and Vertical Combustion

The TGA and DTG curves of CPCMs are shown in Figure 8. The thermogravimetric data for CPCM are shown in Table 3 (where $T_{onset}$ and $T_{max}$ are defined as 5% weight loss and maximum pyrolysis temperature, respectively). As shown in Figure 8a, The pyrolysis temperatures of CPCMs ranged from 250 °C to 450 °C. Figure 8b illustrates the thermal degradation rate of the samples at different temperatures, from which the temperature at which the maximum thermal degradation rate occurs can be obtained. The thermogravimetric data for the CPCMs are summarized in Table 3. According to the data in Table 3, compared with CPCM1, the onset pyrolysis temperature of RPCM increased by more than 100 °C, the pyrolysis curve shifted to the direction of high temperature, and the range of pyrolysis temperature narrowed at the same time, this suggests that the incorporation of the flame retardant led to a substantial enhancement in the thermal stability of the material. In addition, the residual carbon content after pyrolysis of CPCM1 in TGA is almost zero, indicating that the carbonization ability of CPCM without added flame retardant is poor. With the increase of the proportion of added flame retardant, the residual carbon content gradually increases up to 24.83%, which has superior carbonation ability. It can be judged that IFR enhances the carbonization ability of CPCM, and the dense carbon layer prevents the entry of external air and heat, thus improving thermal stability and significantly increasing the onset of degradation temperature. Therefore, the CS/APP/PA-Ba flame retardant system can significantly enhance the fire resistance properties of CPCMs.

The leakage of the sample will affect the stability of the CPCM thermal storage module and the safety of BTMS. The samples were placed in a drying oven at 70 °C, and the leakage experiments were carried out for 1, 2, 3, 4, 5, and 6 h sequentially, and the weights were recorded. The image of the leakage rate is shown in Figure 9, and the mass loss of the sample without IFR was larger, with a leakage rate of 9.98%. The leakage rate of the samples with different ratios of CS/APP/PA-Ba flame retardant system was in the range of 1.7~2.8%, and the quality retention rate was significantly improved. The results show that the addition of IFR can effectively improve the quality stability of PCMs and reduce the leakage of organic matter.

The vertical burning results of CPCMs with varying ratios of flame retardants are listed in Table 4. Vertical burning is used to evaluate the self-extinguishing ability of the materials after ignition and to judge their fire resistance grade. According to the self-extinguishing time after burning, it is divided into four grades: V-0, V-1, V-2, and ungraded, with V-0 representing the highest fire resistance grade. When the amount of IFR added was increased to 15%, all samples achieved the top V-0 rating in vertical combustion grades; this can be attributed to enough IFR, which promptly forms a dense carbon layer on the material’s surface, effectively insulating it from oxygen and heat and thereby achieving flame retardancy. As depicted in Figure 10, the vertical combustion details of the samples are illustrated, such as Figure 10a CPCM1 without adding flame retardant; after 10 s of spraying, the fire continued to burn, the PEG quickly melted and large pieces of drops, indicating that the material is thermally unstable, the surface of the carbon layer is basically no generation of no flame-retardant effect. The fact that the material was extinguished in 32 s does not mean that it can self-extinguish, but due to the inability to produce a dense carbon layer, the material melted and dripped. If applied in battery modules, the melting of the material by heat will only cause larger-scale heat spread and thermal runaway accidents. As shown in Figure 10b, the suboptimal flame-retardant effect of RPCM1 was attributed to the inadequate amount of flame retardant, sluggish carbon layer growth, and subpar carbon layer quality. Therefore, after the completion of the second ignition for 60 s, the specimen strip is still burning and cannot pass the vertical combustion test. The reduction in self-extinguishing time observed in the experiment can be directly related to the flame-retardant additives. When 30% IFR is added, the material is extinguished at 1 s. It can be inferred that when the proportion of IFR is higher, a dense carbon layer can be grown faster to isolate the oxygen and heat to achieve a better flame-retardant effect. However, it is not only better to add more IFR in the material but also to consider the effect on latent heat and the practical effect of applying it to the thermal management of the battery. Compared to other flame retardant PCMs, the novel flame retardant solid–solid PCM [20] employs microencapsulated APP as the flame retardant. The material achieved a V-0 rating in vertical burning tests with a flame-retardant content of 19%. Additionally, as mentioned earlier, a flame retardant PCM was prepared to investigate the synergistic flame retardant effect of APP and RP [35] and its application in thermal management. The optimal flame-retardant performance was observed when the mass fraction of APP and RP was 33%. In this study, an environmentally friendly IFR system based on APP was used to reduce the IFR content in the phase change material to 15% by mass, achieving a V-0 rating in vertical burning tests.

### 3.3. Battery Module Heat Dissipation Performance

#### 3.3.1. Cell

Experiments were carried out to further investigate the impact of varying flame-retardant contents in RPCMs as heat dissipation media for single-cell thermal management and to validate the temperature control performance of the materials. The experiments were carried out in a constant temperature incubator at 25 °C, and the thermal management effects of the five RPCMs and the naturally cooled Group Blank without CPCM were tested, respectively. Figure 11 shows the variation of cell surface temperature with time at two discharge rates, 2C and 3C. In Figure 11a, at 2C, the battery temperature rises sharply during the constant-current discharge phase, and the maximum temperature of group Blank reaches 53.67 °C. The maximum temperatures of the RPCM-cooled batteries are 44.56 °C, 45.80 °C, 47.1 °C, 45.71 °C, and 49.98 °C in the order of the maximum temperature of the RPCM-cooled batteries. The material RPCM1 has the best cooling effect, with the surface temperature of the battery reduced by 9.11 °C. The main reason is that RPCM1 has high latent heat and thermal conductivity, enabling it to rapidly absorb and transfer heat. In contrast, when air is used as the medium, its low thermal conductivity leads to inefficient heat dissipation. As a result, heat accumulates in the system, causing a rapid increase in temperature. As shown in Figure 11b, the maximum surface temperature of the battery in group Blank reaches 68.15 °C at 3C discharge, while the surface temperature of the battery under RPCM is 58.4~61.69 °C, which is 6.46~9.75 °C lower than that of the LIB in the blank group Blank. The highest temperature recorded for the battery with RPCM cooling is 44.56 °C, 45.80 °C, 47.1 °C, 45.71 °C, and 49.98 °C, respectively. Compared with natural cooling, the use of RPCM noticeably reduces the surface temperature of the battery module. RPCM can absorb heat generated by cells during thermal management, effectively controlling the temperature within a safe operating range of 60 °C [10]. It serves as a temperature buffer.

#### 3.3.2. Battery Module

The effectiveness of RPCM in battery module thermal management will be further investigated in the upcoming research. Observing Figure 12a, it can be found that the peak temperature of the battery module under natural air-cooling conditions at 2C is 59.01 °C. After using RPCM, the surface temperature drops to 54.24~56.47 °C, which is a temperature reduction of 2.54~4.76 °C. At 3C, as depicted in Figure 12b, the temperature of the battery without RPCM rises sharply, and the peak surface temperature of the battery module reaches 81.53 °C. After using RPCM, the temperature drops to 62.82 °C, 64.45 °C, 66.19 °C, 65.41 °C, and 69.59 °C, with the temperature reduction ranging from 11.94 to 18.71 °C, and the temperature is lowered by a maximum of 22.9%, respectively. The temperature reduction effect was remarkable. The results demonstrate that the RPCM can buffer rapid temperature changes, absorb heat during battery temperature increases, release heat during temperature decreases, and exhibit good temperature control performance. Compared to other PCMs used in thermal management in the field, the cooling effect of this material remains at an above-average level. A PCM synthesized by incorporating EG, boron nitride (BN), and silicone rubber into paraffin was employed for thermal management [47]. During 3C discharge, the maximum temperature of the battery was 14.6 °C lower than that of natural cooling. Furthermore, a flexible PCM composed of styrene-butadiene-styrene (SBS), paraffin, and aluminum nitride (AlN) was utilized for BTMS [48]. At 2C discharge, the surface temperature of the battery decreased by 5.8 °C compared to natural air cooling, while at 3C discharge, the temperature dropped by 6.5 °C. By conducting experimental analysis, RPCM1 and RPCM2 have the best cooling effect. Adding high thermal conductivity material inside the flame retardant PCM can improve the heat dissipation ability while ensuring the flame-retardant performance. Considering its flame-retardant performance, due to the excellent flame retardancy of RPCM2, which achieves a V0 rating, its application in battery thermal management systems is highly advantageous for overall safety during operation.

A temperature variation exists among the cells within the battery module, and it is essential to ensure that the maximum temperature difference does not exceed 5 °C; exceeding this limit is detrimental to the proper functioning of the battery module, jeopardizing both safety and the overall module lifespan. The temperature difference curves of the battery module are shown in Figure 13, and the temperature difference under 2C conditions is 7.18 °C without using RPCM. After using RPCM, the temperature difference was reduced to the range of 2.24 to 3.09 °C. In the 3C condition, the temperature difference was 7.18 °C without RPCM. In the 3C condition, the temperature difference without RPCM was larger, 11.48 °C and the surface temperature difference after using RPCM decreased significantly, 5.36 °C for the RPCM5 group, and the temperature difference for the rest of the groups ranged from 4.28 to 4.94 °C, which was less than 5 °C, with the highest temperature difference reduced by 62.7%. It can be inferred that the use of RCPCM can significantly reduce the temperature variation observed within the lithium-ion battery module, which can be controlled within 5 °C to improve its safety and service life.

## 4. Conclusions

In this study, a novel RPCM was prepared using PEG and bio-based flame retardant. The main conclusions are as follows:(1)After the addition of MWCNTs and EG, the material exhibits a significant increase of 558% in thermal conductivity compared to pure PEG. The addition of CS/APP/PA-Ba flame retardants significantly increased the residual carbon content and flame retardancy of the material while reducing organic leakage, effectively enhancing the thermal stability of PCM.(2)When the flame-retardant content was 15%, the material RPCM2 achieved a V-0 flame retardant rating, with a thermal conductivity of 2.04 W/m·K and a latent heat value of 151.58 W/g. Therefore, RPCM2 not only maintains excellent thermodynamic performance but also exhibits outstanding flame retardancy and good thermal stability.(3)Compared to natural cooling, the use of the novel RPCM as a heat dissipation medium in thermal management demonstrated a significant cooling effect. In a 3C cycle, the battery module experienced a reduction of 18.71 °C in its maximum temperature, and the temperature difference decreased by 62.7% compared to natural cooling.

The prepared RPCM in this study provides a reference for the application of bio-based flame retardants in CPCM. However, further research is required for the practical application of these materials.

## Figures and Tables

**Figure 1 polymers-15-04450-f001:**
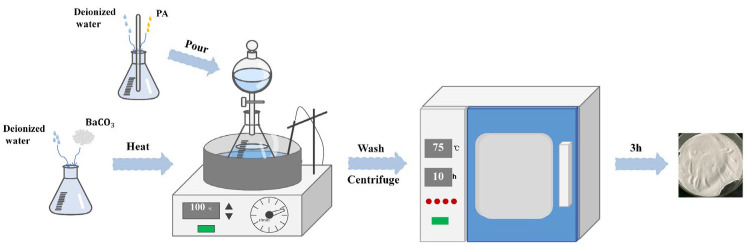
The preparation process of PA-Ba.

**Figure 2 polymers-15-04450-f002:**
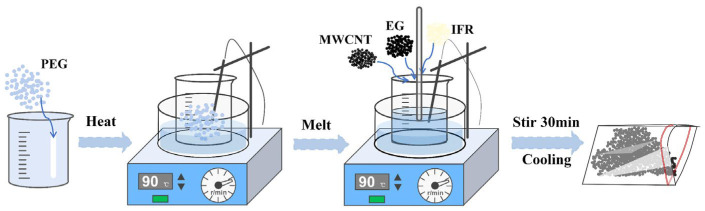
Preparation of flame retardant CPCMs.

**Figure 3 polymers-15-04450-f003:**
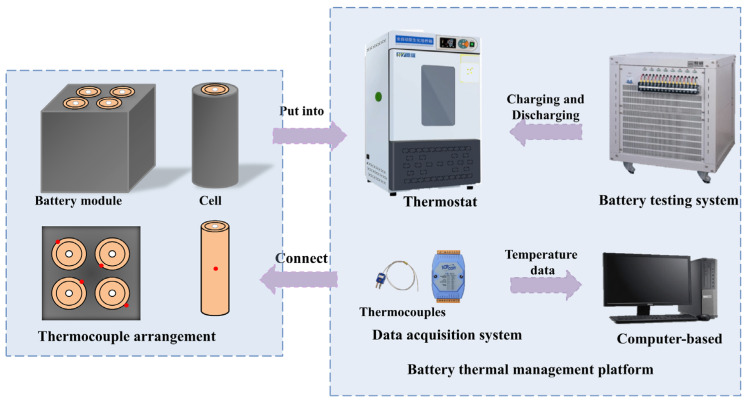
Battery thermal management platform setup schematic.

**Figure 4 polymers-15-04450-f004:**
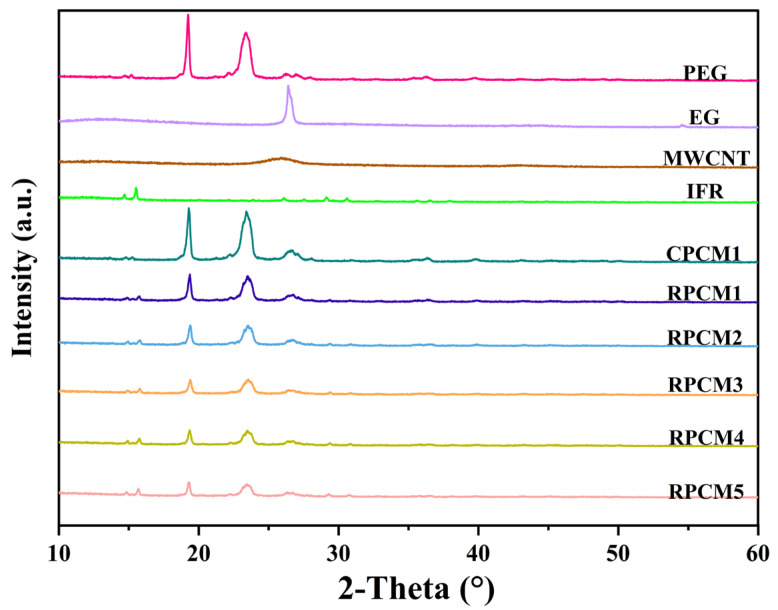
XRD graph.

**Figure 5 polymers-15-04450-f005:**
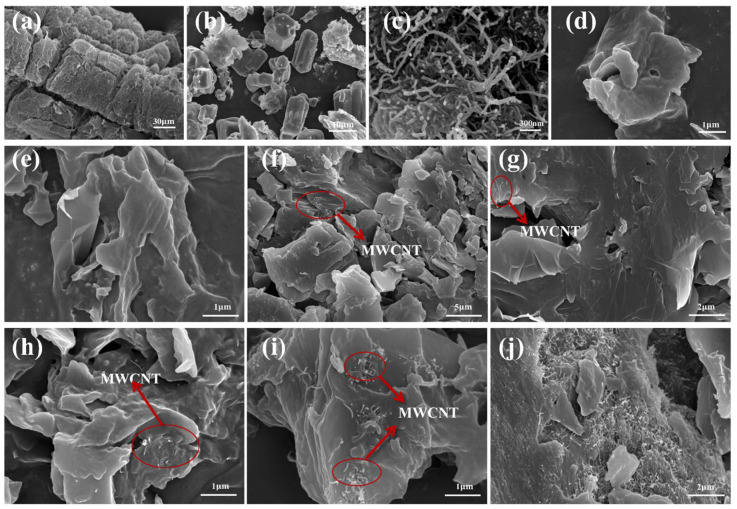
SEM morphology of (**a**) the worm-like structure of EG; (**b**) PA-Ba; (**c**) MWCNT; (**d**) PEG; (**e**–**j**) CPCM1, RPCM1-5.

**Figure 6 polymers-15-04450-f006:**
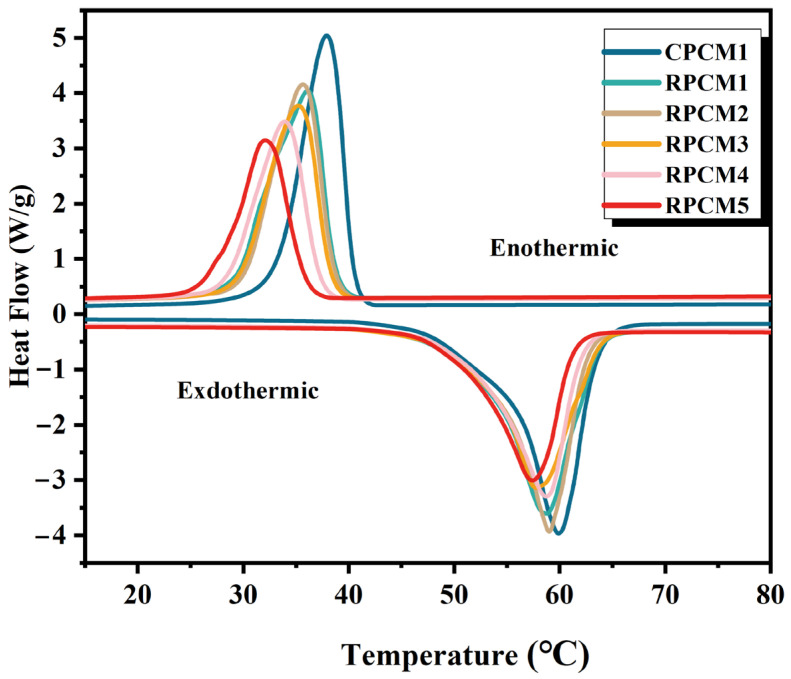
DSC curve of each sample.

**Figure 7 polymers-15-04450-f007:**
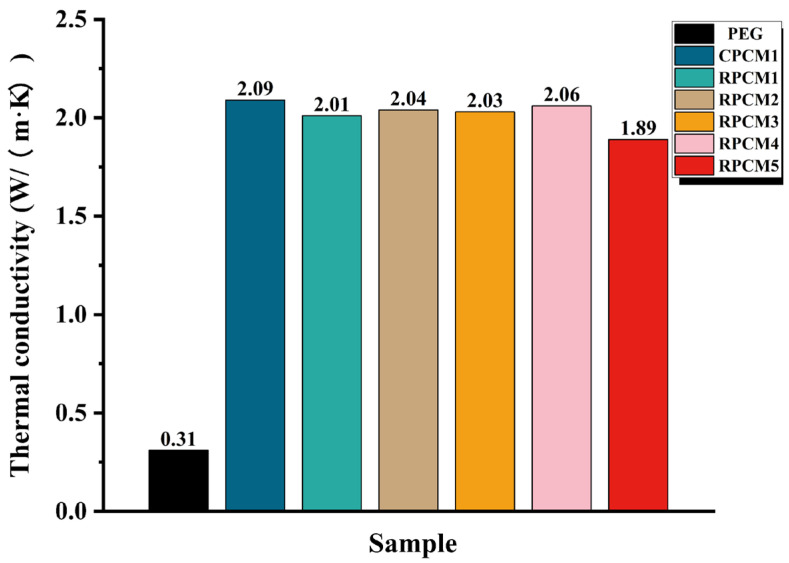
Thermal conductivity of each sample.

**Figure 8 polymers-15-04450-f008:**
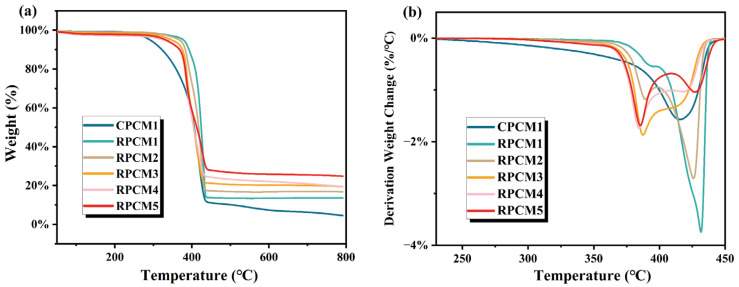
(**a**) TGA and (**b**) DTG curves for different samples.

**Figure 9 polymers-15-04450-f009:**
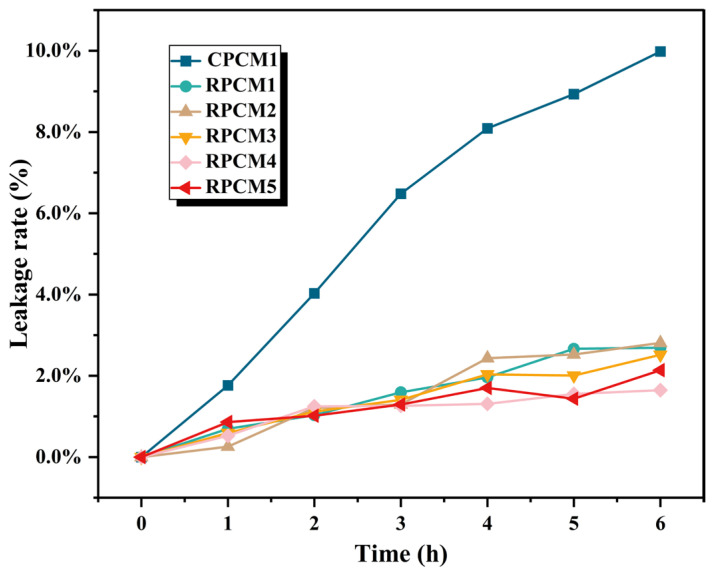
Leakage rate graph.

**Figure 10 polymers-15-04450-f010:**
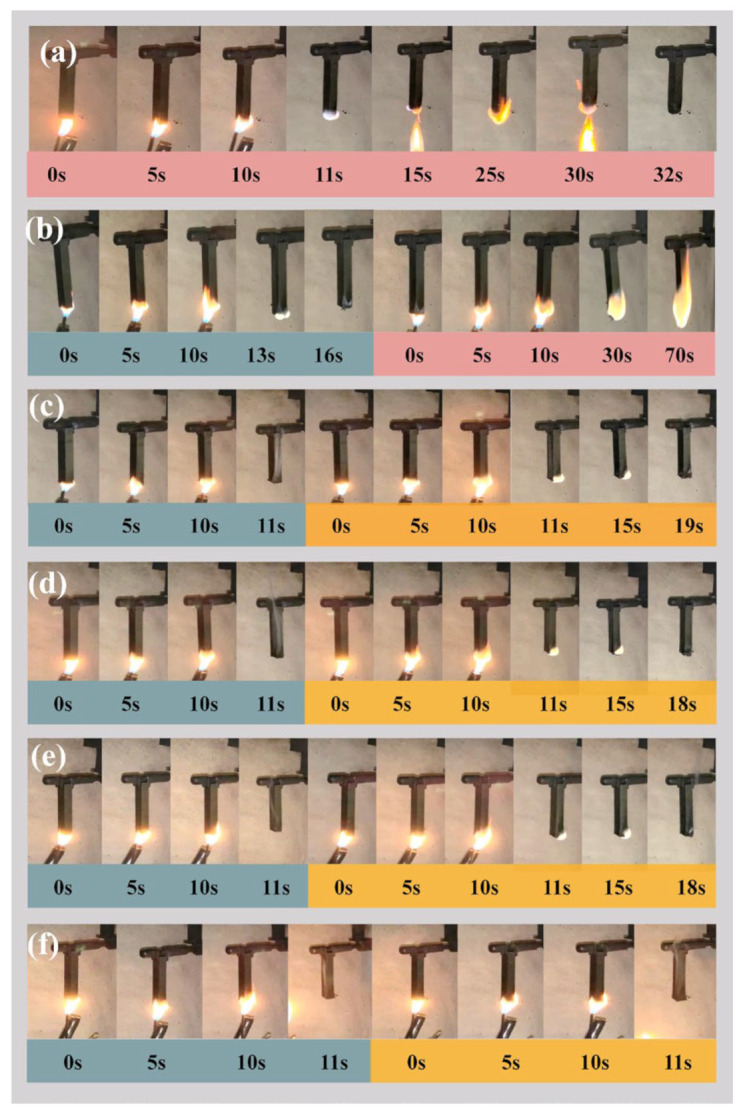
(**a**) CPCM1 combustion process; (**b**) RPCM1 combustion process; (**c**) RPCM2 combustion process; (**d**) RPCM3 combustion process; (**e**) RPCM4 combustion process; (**f**) RPCM5 combustion process.

**Figure 11 polymers-15-04450-f011:**
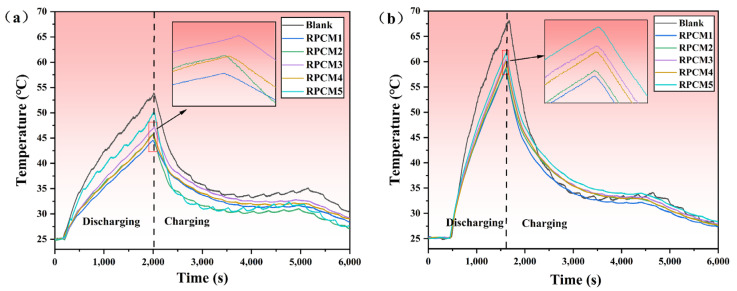
Cell temperature curves of RPCMs and blank under charging rates of (**a**) 2C; (**b**) 3C.

**Figure 12 polymers-15-04450-f012:**
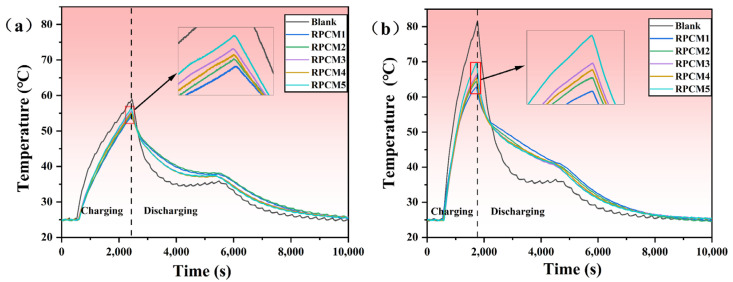
Battery module temperature profiles of RPCMs and blank under charging rates of (**a**) 2C; (**b**) 3 C.

**Figure 13 polymers-15-04450-f013:**
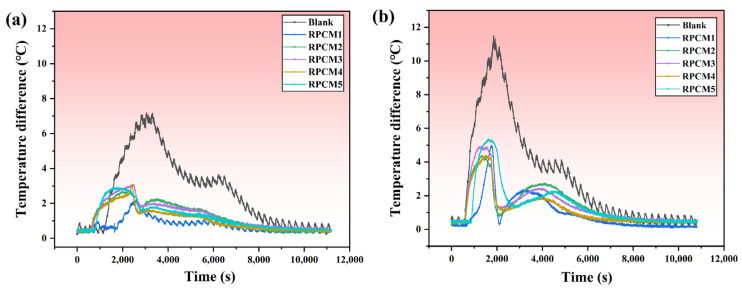
Battery module temperature difference profiles of RPCMs and blank under charging rates of (**a**) 2C; (**b**) 3C.

**Table 1 polymers-15-04450-t001:** Composition of CPCMs.

Samples	PEG (wt%)	EG + MWCNT (wt%)	IFR (wt%)
CPCM1	93	5 + 2	0
RPCM1	83	5 + 2	10
RPCM2	78	5 + 2	15
RPCM3	73	5 + 2	20
RPCM4	68	5 + 2	25
RPCM5	63	5 + 2	30

**Table 2 polymers-15-04450-t002:** Thermophysical parameters of PCMs.

Samples	$T_{s\text{-}l}$ (°C)	Latent Heat of Phase Change Δ*H* (w/g)
PEG	57.32	183.73
CPCM1	54.62	166.51
RPCM1	52.89	157.55
RPCM2	53.53	151.58
RPCM3	52.93	141.24
RPCM4	51.99	131.78
RPCM5	51.09	119.45

**Table 3 polymers-15-04450-t003:** Specific thermogravimetric parameters of CPCM.

Samples	$T_{\mathrm{onset}}$ (°C)	$T_{\max}$ (°C)
CPCM1	252.54	415.77
RPCM1	375.59	429.78
RPCM2	368.24	424.83
RPCM3	362.19	388.20
RPCM4	359.52	385.01
RPCM5	350.76	385.56

**Table 4 polymers-15-04450-t004:** Vertical combustion rating for each sample.

Samples	CPCM1	RPCM1	RPCM2	RPCM3	RPCM4	RPCM5
IFR (%)	0	10	15	20	25	30
UL-94	/	/	V0	V0	V0	V0

## Data Availability

Data are contained within the article.

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
