# Peer review of "Application of Polyethylene Glycol-Based Flame-Retardant Phase Change Materials in the Thermal Management of Lithium-Ion Batteries"

_polymers, 2023, doi:10.3390/polym15224450_

Round 1
Reviewer 1 Report
Comments and Suggestions for Authors
The authors introduced a PEG-based composite material for thermal management applications in LIBs. The background has been clearly introduced, and abundant literatures are summarized to point out the significance and novelty of this work. This material has been demonstrated to be successful in having significantly higher thermal conductivity and flame retardancy compared with PEG. Additionally, some preliminary testing of thermal management on battery cells and modules was used to demonstrate the potential of being applied in real LIB products. Therefore, I am happy to recommend this article being published after the following minor revisions:
1) Table 1: Please clarify the weight percentage in the caption.
2) In Figure 4, most of the peaks showed up below 60 degree. Please zoom in 10-60 degree and no need to show the 60-80 portion.
3) Figure 5: The scale bars are too small to be seen clearly. Please trim the figures and add scale bars with corresponding scales manually.
4) Figure 6: Please clarify if positive heat flow is exothermic or endothermic in captions or texts.
5) Table 2 is not composition of CPCMs, please revise.
6) Caption of Figure 11 should be a complete sentence: cell temperature curve of RPCMs and blank under charging rate of (a) 2C and (b) 3C.
7) 3.3.2 should have title of "battery module" instead of "cells".
Comments on the Quality of English LanguageThe title needs small corrections on grammar, e.g., "Glycol-Based", and "Flame-Retardant".
Reviewer 2 Report
Comments and Suggestions for Authors
The manuscript" Application of Polyethylene Glycol based Flame Retardant Phase Change Materials in Thermal Management of Lithium-Ion Batteries" shows some interesting concepts using PEG based flame retardants. The manuscript needs revision.
1. The use of different flame retardants are analyzed but there need to be especially since EG and MWCNT added the characterization of such, using Raman spectroscopy and for PEG components FTIR. Please add those
2. The main text is difficult to read due to the many abbreviation used in the manuscript. The reviewer suggest either add a Table in supplementary where all abbreviations listed or refine those several times in the main test.
3. The main concern are the missing discussion as in the result part there must be references and discussion provided of other works made in the field. Please include those or provide an extra section called discussion.
4. The flame retardant itself shows some small effect in reducing temperature but how does such behave as filler in membranes of actually Li batteries. Did the authors perform such test?
5. There certain parts needs better explained as partly given in the introduction it would be good having a Table of comparison to other flame retardants made in the field. Please add such before conclusion that the reader can evaluate how well your flame retardants are in comparison to other made before.
6. There some minor mistakes and some abbreviations not defined such as PA, CS. Also there a double point shown on page 3, line 137. Please check your manuscript with all minor spell checking
Comments on the Quality of English Language
The English is fine with minor spell checking needed
Round 2
Reviewer 2 Report
Comments and Suggestions for Authors
The authors made revision and answered all open question.
Author Response
We sincerely appreciate your help and approval of our modifications in the manuscript.